# Phosphate and Inflammation in Health and Kidney Disease

**DOI:** 10.3390/ijms27010408

**Published:** 2025-12-30

**Authors:** Carlos Novillo-Sarmiento, Raquel M. García-Sáez, Antonio Rivas-Domínguez, Ana Torralba-Duque, Cristian Rodelo-Haad, María E. Rodríguez-Ortiz, Juan R. Muñoz-Castañeda, M. Victoria Pendón-Ruiz de Mier

**Affiliations:** 1Unidad de Gestión Clínica Nefrología, Reina Sofia University Hospital, 14004 Cordoba, Spain; carlosnov13@hotmail.com (C.N.-S.); anatduque@gmail.com (A.T.-D.); juanr.munoz.exts@juntadeandalucia.es (J.R.M.-C.); mvictoriaprm@gmail.com (M.V.P.-R.d.M.); 2Instituto Maimónides de Investigación Biomédica de Córdoba (IMIBIC), Reina Sofia University Hospital, Department of Medicine, University of Cordoba, 14004 Cordoba, Spain; raquel.garcia@imibic.org (R.M.G.-S.); antonio.rivas@imibic.org (A.R.-D.); marien_rguez@hotmail.com (M.E.R.-O.); 3Redes de Investigación Cooperativa Orientadas a Resultados en Salud (RICORS), Instituto de Salud Carlos III, RD24/0004/0004, 28029 Madrid, Spain

**Keywords:** phosphate overload, oxidative stress, FGF23–Klotho axis, vascular calcification, chronic kidney disease, magnesium, NOX/ROS signaling, SIRT1

## Abstract

Phosphate is emerging as an active mediator of oxidative stress and vascular injury in chronic kidney disease (CKD). This emerging pathophysiological framework, referred to as “Phosphatopathy”, describes the systemic syndrome driven by chronic phosphate overload and characterized by oxidative stress, inflammation, endothelial dysfunction, vascular calcification, cellular senescence, and metabolic imbalance. Beyond being a biochemical marker, phosphate overload triggers NOX-derived reactive oxygen species (ROS), activates Wnt/β-catenin and TGF-β signaling, and disrupts the FGF23–Klotho axis, promoting endothelial dysfunction, vascular calcification, and left ventricular hypertrophy (LVH). These pathways converge with systemic inflammation and energy imbalance, contributing to the malnutrition–inflammation–atherosclerosis (MIA) syndrome. Experimental and clinical data reveal that the phosphate/urinary urea nitrogen (P/UUN) ratio is a sensitive biomarker of inorganic phosphate load, while emerging regulators such as microRNA-125b and calciprotein particles integrate phosphate-driven oxidative and inflammatory responses. Therapeutic strategies targeting phosphate burden—rather than serum phosphate alone—include dietary restriction of inorganic phosphate, non-calcium binders, magnesium and zinc supplementation, and activation of important pathways related to the activation of antioxidant defense such as AMP-activated protein kinase (AMPK) and SIRT1. This integrative framework redefines phosphate as a modifiable upstream trigger of oxidative and metabolic stress in CKD. Controlling phosphate load and redox imbalance emerges as a convergent strategy to prevent vascular calcification, improve arterial stiffness, and reduce cardiovascular risk through personalized, mechanism-based interventions.

## 1. Introduction

Phosphate is an essential mineral for both cellular and systemic homeostasis, playing a key role in ATP production, intracellular signaling pathways, and hydroxyapatite formation in bone. Under physiological conditions, serum phosphate levels are maintained within a narrow range (2.5–4.5 mg/dL) through a complex balance among intestinal absorption, bone exchange, and renal excretion. This regulation depends on a coordinated endocrine axis involving parathyroid hormone (PTH), active vitamin D 1,25 di-hydroxy vitamin D), fibroblast growth factor 23 (FGF23), and the α-Klotho protein (referred to as Klotho throughout the text) [1].

In CKD, the progressive decline in the estimated glomerular filtration rate (eGFR) reduces phosphate excretory capacity. Initially, compensatory increases in FGF23 and PTH enhance phosphaturia and decrease tubular phosphate reabsorption. However, secondary hyperparathyroidism and excessive FGF23 secretion are linked to deleterious effects, including LVH, systemic inflammation, and premature mortality [2,3]. When eGFR falls below 30 mL/min/1.73 m^2^, these adaptive mechanisms fail, leading to overt hyperphosphatemia and the development of the CKD–mineral and bone disorder syndrome [4].

Beyond elevated serum levels, phosphate overload has emerged as a key driver of metabolic and vascular injury. Chronic phosphate imbalance not only reflects declining renal function but also contributes directly to inflammation, oxidative stress, and tissue remodeling. These deleterious effects extend to the endothelium, myocardium, and kidney, and have been recognized as major contributors to the high cardiovascular burden of CKD [5].

Systemic phosphate overload also induces inflammation and malnutrition, generating a pathological feedback loop known as the MIA syndrome. It is characterized by elevated pro-inflammatory cytokines (TNF-α, IL-6), decreased serum albumin, and loss of muscle mass—factors that substantially increase cardiovascular morbidity and mortality in CKD [6,7]. The impact of oxidative stress extends beyond the vasculature as recent studies have shown that excessive ROS impair hematopoietic stem cell renewal and differentiation, contributing to erythropoietin-resistant anemia and the immune dysfunction observed in uremic patients [6]. Hyperphosphatemia causes alterations in endothelial cells, including reduced nitric oxide (NO) production associated with oxidative stress, leading to decreased cell viability and increased apoptosis [8]. Moreover, elevated phosphate levels induce endothelial cell senescence through cell cycle arrest, promoting cellular aging rather than apoptosis [9].

Consequently, there is a growing need for biomarkers that reflect phosphate burden and vascular injury earlier than serum phosphate levels become elevated. A simple urinary parameter such as the P/UUN ratio may distinguish between inorganic phosphate intake and natural protein sources, offering a practical tool for nutritional assessment and dietary personalization in CKD [10] and in subjects with normal renal function [11]. Complementarily, microRNAs (miRNAs) have emerged as promising tools. Specifically, miR-125b acts as a negative regulator of osteogenic transdifferentiation in vascular smooth muscle cells (VSMCs). Low circulating miR-125b levels correlate with more severe vascular calcification and predict its progression in end-stage kidney disease, regardless/independently of phosphate, FGF23, PTH, or vitamin D status [6]. This supports the possible role of miR-125b as a non-invasive biomarker with potential translational into clinical practice [12].

Overall, positive phosphate balance should be understood not merely as a metabolite to be controlled, but as a true upstream trigger of oxidative stress and inflammation [13] that interacts with the FGF23–Klotho–PTH axis and is reflected early in urinary biomarkers such as P/UUN [10]. These processes ultimately drive cardiovascular complications and premature death.

The present review summarizes the role of phosphate excess on the promotion of systemic inflammation, the MIA syndrome, cardiac and vascular remodeling and the new link between phosphate alterations and energy metabolism. Thus, the term Phosphatopathy could be assumed as a systemic syndrome of phosphate-driven metabolic, vascular and cellular dysfunction, including endothelial injury, vascular calcification, cellular senescence and organ aging, attributable to chronic phosphate overload [14]. Despite not being an established diagnostic category, the term “phosphatopathy” is used here as a conceptual framework to integrate the multisystem effects of chronic phosphate overload. In this context, we review the main factors and mechanisms that contribute to the development of phosphatopathies. Although this review focuses primarily on the toxic effects of phosphate overload, it is important to recognize that phosphate-related pathophysiology is bidirectional. Several large population-based studies have demonstrated a U-shaped association between serum phosphate levels and mortality, indicating that both excessively high and low phosphate concentrations may be detrimental to health. Hypophosphatemia may occur as a consequence of increased renal phosphate wasting, often mediated by elevated FGF23, as seen in hereditary hypophosphatemic disorders (e.g., XLH), tumor-induced osteomalacia, or acquired tubular dysfunction. Renal tubular disorders such as Fanconi syndrome or proximal tubular injury can further promote phosphate loss, contributing to muscle weakness, impaired bone mineralization, and, in severe cases, renal dysfunction. These observations highlight that maintaining phosphate balance within a narrow physiological range is essential and that disturbances on either side of this spectrum can exert clinically significant systemic effects [15].

## 2. Phosphate Regulation (FGF23, Klotho, PTH)

Phosphate metabolism depends on a delicate equilibrium among intestinal absorption, bone utilization, and renal excretion. Intestinal absorption occurs primarily via the NPT2b cotransporter, whose expression is regulated by dietary phosphate and active vitamin D [16,17]. Bone serves as a dynamic phosphate reservoir, exchanging phosphate according to metabolic demands. The kidney, specifically the proximal tubule, is the main regulatory organ, reabsorbing about 80% of filtered phosphate under normal conditions [10,18].

### The FGF23–Klotho–PTH Axis Constitutes the Central Endocrine System Controlling Phosphate Homeostasis

FGF23, secreted mainly by osteocytes and osteoblasts in response to phosphate loading, reduces tubular phosphate reabsorption (by inhibiting NPT2a and NPT2c cotransporters in the proximal tubule) and suppresses renal 1,25 di-hydroxy vitamin D synthesis, thereby limiting intestinal phosphate absorption. Excessive FGF23 has been linked to LVH, inflammation, immunosuppression, anemia, bone abnormalities and the transition of VSMC from contractile to synthetic phenotype [2,17,19] (Figure 1).

Klotho, expressed in the distal renal tubule, acts as co-receptor for FGF23 signaling. Beyond this, it exerts independent antioxidant and anti-fibrotic effects, regulating calcium homeostasis and protecting against vascular and renal aging [20,21]. Loss of Klotho expression in CKD promotes activation of profibrotic pathways such as Wnt/β-catenin and TGF-β [20]. Klotho binds to different Wnt ligands halting its activation [21,22].

PTH responds to elevated phosphate and reduced ionized calcium levels by stimulating renal phosphate excretion and enhancing bone resorption. In CKD, secondary hyperparathyroidism arises from a combination of hypocalcemia, vitamin D deficiency, and phosphate retention. Conversely, phosphate acts on parathyroid calcium-sensing receptor (CaSR) to directly stimulate PTH secretion [18,23].

The interplay of these three regulators determines phosphate balance and directly connects mineral metabolism with cardiovascular pathophysiology. Elevated FGF23 and reduced Klotho levels have been linked to arterial stiffness, vascular calcification, and endothelial dysfunction [17,24]. Indeed, clinical studies have demonstrated that high circulating FGF23 independently predicts mortality in hemodialysis patients, even after adjusting for serum phosphate [16] (Figure 1).

Thus, chronic phosphate overload and disruption of the FGF23–Klotho–PTH axis form a central node linking mineral metabolism with cardiovascular injury.

## 3. Role of Phosphate Excess on Promotion of Systemic Inflammation

Phosphate exerts essential physiological. However, once extracellular phosphate exceeds renal excretory capacity, it transitions from a homeostatic substrate to a potent inducer of oxidative stress and inflammation. In this section, we focus specifically on the molecular pathways through which phosphate overload triggers inflammatory signaling [1,17]. Although phosphate can also trigger inflammatory signalling in other organs, including the liver or lung, these mechanisms are not addressed here, as this section focuses on pathways relevant to CKD and cardiovascular injury.

Experimental and preclinical data have elucidated the molecular basis of phosphate-induced inflammation and oxidative stress. In VSMC, high extracellular phosphate concentrations (>3 mM) promote phosphate influx through the PiT-1 cotransporter [25], leading to ROS generation, NF-κB activation, and the expression of osteogenic and inflammatory mediators such as Runx2, osteocalcin, IL-6, and TNF-α [20] (Figure 2). In uremic animal models, high-phosphate diets cause vascular calcium-phosphate deposition, activation of NADPH oxidase (NOX4), and accumulation of oxidative markers such as 8-hydroxy-2′-deoxyguanosine (8-OHdG) [6,26,27]. Similarly, phosphate-induced activation of Wnt/β-catenin and TGF-β pathways amplifies inflammatory and profibrotic signaling in endothelial and vascular cells [20]. In renal tissue, phosphate loading triggers tubular injury, Klotho suppression, and mitochondrial oxidative stress, while in HEK-293 cells it directly increases ROS production in proportion to extracellular phosphate concentrations. Collectively, these data identify phosphate overload as a cellular stressor that disrupts redox homeostasis and activates pro-inflammatory signaling across multiple tissues [8,10] (Figure 2).

Clinically, elevated phosphate levels are associated with increased inflammatory biomarkers, vascular stiffness, and endothelial dysfunction, even in individuals with normal renal function. Epidemiological and metabolic studies demonstrate that highly absorbable inorganic phosphate additives in processed foods elicit postprandial surges in phosphate and FGF23, which promote endothelial oxidative stress and reduce nitric oxide bioavailability [18,28].

At the molecular level, phosphate excess activates multiple ROS-generating systems, including mitochondria, NADPH oxidase (especially NOX4), and uncoupled endothelial nitric oxide synthase, creating a state of oxidative distress—a sustained imbalance that damages DNA, proteins, and lipids beyond physiological redox signaling [13,29]. The ROS–NF-κB axis acts as a self-perpetuating amplifier of inflammation, establishing a positive feedback loop that accelerates CKD progression, endothelial dysfunction, and vascular remodeling [13,25,29].

Phosphate-induced oxidative stress is not limited to the cardiovascular system. In bone marrow, excessive ROS impairs hematopoietic stem cell renewal, promoting senescence and apoptosis, which contribute to erythropoietin-resistant anemia (6). Moreover, in metabolic disorders such as type 2 diabetes, high phosphate levels enhance mitochondrial ROS and NOX activity, linking phosphate toxicity with insulin resistance and vascular injury [13,30].

Together, these findings highlight phosphate overload as a potent upstream driver of oxidative stress and inflammation, integrating mineral metabolism with vascular, hematopoietic, and metabolic dysfunction. The key molecular pathways described in this section, including PiT-1–mediated phosphate influx, NOX4-derived ROS, NF-κB activation, mitochondrial dysfunction, and early osteogenic signaling—are summarized in Figure 2 and constitute the mechanistic foundation for the MIA syndrome (Section 4) and vascular calcification (Section 5).

## 4. The MIA Syndrome (Malnutrition–Inflammation–Atherosclerosis)

The MIA syndrome is a key pathogenic triad in CKD, arising from the interplay of phosphate overload, chronic oxidative stress, and systemic inflammation. This axis decisively contributes to higher cardiovascular morbidity and mortality in advanced CKD and dialysis [31].

### 4.1. Molecular and Pathophysiological Basis

Accumulation of phosphate and uremic toxins activates pro-inflammatory and pro-oxidant pathways across multiple tissues. In animal models, high-phosphate diets (1.2%) increase TNF-α, IL-6, and C-reactive protein (CRP) and lead to weight loss, hypoalbuminemia, and muscle atrophy [6]. This phenotype coincides with higher 8-OhdG, a marker of oxidative DNA damage, and with NOX4 overexpression and NF-κB activation [13].

As detailed in Section 3 and illustrated in Figure 2, phosphate-induced oxidative stress establishes a systemic pro-inflammatory milieu. Building on these upstream events, phosphate overload disrupts mitochondrial oxidative phosphorylation and impairs cellular energy sensing by inhibiting AMPK and SIRT1, two key regulators of mitochondrial biogenesis, redox balance, and metabolic homeostasis [32]. In CKD, dysfunction of these pathways perpetuates a low–energy-expenditure state and chronic inflammation.

These energetic disturbances contribute to the low-energy-expenditure state characteristic of CKD and amplify chronic inflammation. In parallel, the oxidative environment described in Section 3 adversely affects hematopoiesis, reducing hematopoietic stem cell self-renewal and impairing erythroid differentiation, which promotes the erythropoietin-resistant anemia observed in advanced CKD [33]. This inflammatory anemia worsens tissue hypoxia and ROS generation, closing a vicious cycle of oxidative injury and metabolic dysfunction.

### 4.2. Interaction with Vascular Inflammation and Atherosclerosis

Systemic inflammation driven by phosphate and ROS promotes endothelial dysfunction by reducing bioavailable nitric oxide and upregulating adhesion molecules such as VCAM-1 and ICAM-1, which enhance leukocyte adhesion. Within the arterial wall, recruited monocytes and lymphocytes activate pro-inflammatory macrophages that produce ROS and cytokines, accelerating atherogenesis [34].

In parallel, increased FGF23 and loss of Klotho impair endothelial function. Through FGFR4, FGF23 activates the PLCγ–calcineurin–NFAT pathway, leading to myocardial hypertrophy and vascular dysfunction [2], while Klotho deficiency enhances Wnt/β-catenin and TGF-β signaling, favoring vascular fibrosis [20,35]. These structural changes consolidate the arterial stiffness and calcification described above.

### 4.3. Clinical and Translational Perspective

Clinically, MIA portends poor outcomes. Its presence predicts lower survival, higher hospitalization rates, and loss of muscle mass [31]. Hypoalbuminemia (<3.8 g/dL) and elevated high-sensitivity CRP (>3 mg/L) are common markers [11]. Observational studies show that persistent inflammation correlates with progression of vascular calcification, arterial stiffness, and cardiovascular mortality, even after adjusting for serum phosphate and calcium [25].

From a translational standpoint, several biomarkers can help gauge inflammatory and oxidative burden:Elevated FGF23 levels predict cardiovascular mortality and CKD progression, although direct therapeutic modulation remains controversial [17].Lower serum levels miR-125b reflect osteogenic activation and VC progression, integrating phosphate–inflammation–endothelium signaling [7].P/UUN ratio represents a newly described urinary metric that identifies excessive dietary inorganic phosphate, useful for individualized nutritional stratification and counseling [10,11].

Regarding possible therapeutic strategies, managing MIA requires a multifaceted approach (Figure 3):The first and pivotal therapeutic strategy is phosphate control. Dietary restriction in addition to non-calcium based binders to reduce both phosphate load and systemic inflammation are mandatory [6,16]. Caution should be taken with dietary restriction not to promote a worsening in malnutrition. In this sense, the P/UUN ratio may identify those subjects in whom dietary restriction should be directed to reduce processed food [11]. Dietary education helps reduce inorganic phosphate while maintaining adequate protein–calorie intake.As will be described later, magnesium and zinc supplementation are likely to improve insulin sensitivity and decrease ROS, while protecting against vascular calcification [25,36].Antioxidant intervention represents a promising therapeutic approach. Tempol, a superoxide dismutase mimetic, has been shown to reduce NOX4 expression and 8-OHdG levels, attenuating vascular calcification in uremic rats without altering serum phosphate [6,25,37]. Melatonin, with its antioxidant and chronobiological properties, decreases phosphate-induced ROS production in renal cells, protecting mitochondrial function [6]. In HEK-293 cells, high phosphate promoted oxidative stress while melatonin administration reduced ROS generation [5].More recently, some authors abrogate for the modulation of AMPK and SIRT1 by promoting its activation and restore energetic and mitochondrial homeostasis [32,38,39].Other authors have demonstrated that high-protein diets, particularly those based on animal sources, which are also rich in phosphorus, can aggravate chronic inflammation and catabolic processes in patients with CKD. These effects are linked to increases in nitrogenous waste products, urea, and metabolic overload in the kidney, which stimulate inflammatory pathways and accelerate the protein-energy wasting (PEW) syndrome [40]. In renal patients, this persistent inflammation is reflected in elevated biomarkers such as CRP which predict increased morbidity and mortality. Likewise, the same group has reported the renal benefits of low-protein diets [41]. Such diets reduce the burden of metabolic waste products, decrease glomerular hyperfiltration, and slow disease progression, as well as attenuate chronic inflammatory status. The reduction in phosphorus intake that also results from adopting a low-protein diet may likewise contribute to this beneficial dietary effect.

**Figure 3 ijms-27-00408-f003:**
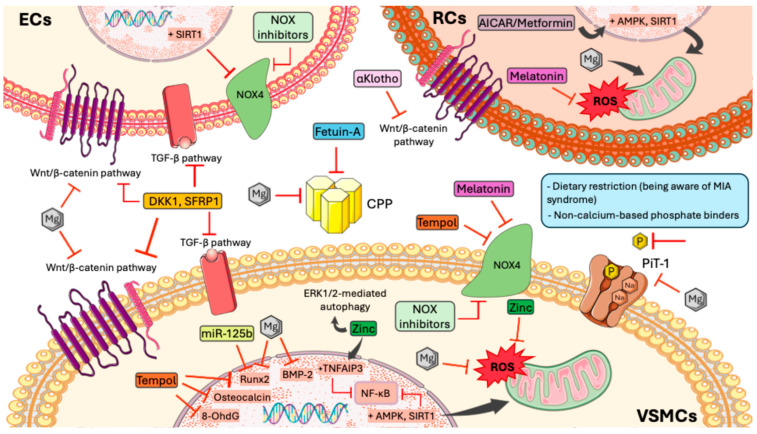
Therapeutic approaches to reduce phosphopaties. Mg^2+^ acts as a pleiotropic modulator exerting multiple protective effects. It mitigates phosphate overload by inhibiting the phosphate transporter PiT-1, thereby reducing ROS generation, inflammation, and fibrosis through modulation of the Wnt/β-catenin signaling pathway and the formation of CPPs. Furthermore, Mg^2+^ downregulates the expression of osteogenic markers such as Runx2 and BMP-2, both implicated in the acquisition of an osteogenic phenotype. In renal cells (RCs), Mg^2+^ improves mitochondrial function and decreases oxidative stress. Together with Fetuin-A, Mg^2+^ limits CPP formation, preventing their deposition in soft tissues. Additionally, miR-125b suppresses Runx2 expression, contributing to the inhibition of vascular calcification. Zinc exerts antioxidative and cytoprotective effects by inhibiting ROS production and activating ERK1/2-mediated autophagy, as well as by inducing TNFAIP3 expression, a known inhibitor of the NF-κB pathway in VSMCs. Similarly, administration of Tempol reduces oxidative markers such as 8-OHdG and NOX4, while attenuating Runx2 and osteocalcin expression in VSMCs [37]. Melatonin also exerts antioxidant effects, reducing NOX4 expression in VSMCs and decreasing ROS production in Rcs. The Wnt/β-catenin and TGF-β signaling pathways are further modulated by DKK1 and SFRP1 in both VSMCs and ECs. In RCs, activation of AMPK and SIRT1 by AICAR or metformin enhances mitochondrial function, whereas in VSMCs these pathways inhibit NF-κB activity. In ECs, SIRT1 also reduces NOX4 expression, thereby attenuating oxidative stress. Additionally, αKlotho modulates the Wnt/β-catenin pathway in RCs, further supporting vascular and renal protection. Complementary therapeutic strategies include dietary phosphate restriction and the use of non–calcium-based phosphate binders to prevent phosphate toxicity and vascular calcification. PiT-1: Phosphate Transporter 1; ROS: Reactive Oxygen Species; CPPs: Calciprotein Particles; Runx2: Runt-related transcription factor 2; BMP-2: Bone Morphogenetic Protein 2; RCs: Renal tubular cells; TNFAIP3: Tumor necrosis factor alpha-induced protein 3; NF-κB: Nuclear Factor kappa B; VSMCs: Vascular Smooth Muscle Cells; 8-OhdG: 8-hydroxy-2′-deoxyguanosine; NOX4: NADPH Oxidase 4; TGF-β: Transforming Growth Factor Beta; DKK1: Dickkopf-1; SFRP1: Secreted Frizzled-Related Protein 1; AMPK: AMP-activated protein kinase; SIRT1: Sirtuin 1; ECs: Endothelial Cells.

### 4.4. Micronutrients and Endogenous Modulators

#### 4.4.1. Magnesium

Magnesium is an essential cation often underestimated in renal and cardiovascular physiology. It acts as a natural calcium antagonist, a universal enzymatic cofactor, and a modulator of intracellular signaling. Under physiological conditions, serum magnesium levels range between 1.7 and 2.4 mg/dL (0.7–1.0 mmol/L), maintained through the coordinated regulation of intestinal absorption, bone storage, and renal excretion—the so-called gut–bone–kidney axis [41]. Approximately 60% of total body magnesium is stored in bone, 20% in muscle, and only about 1% in plasma. Intestinal absorption occurs mainly through paracellular pathways in the small intestine and transcellular routes in the colon via TRPM6 and TRPM7 channels, which also mediate distal tubular reabsorption in the kidney. Claudin-16 and -19 contribute to paracellular transport in the thick ascending limb of Henle’s loop, while CNNM2 and SLC41A3 act as basolateral Mg^2+^/Na^+^ exchangers. Factors such as insulin, EGF, and FGF23 modulate these transporters, which may explain the frequent hypomagnesemia observed in CKD and diabetes [42].

Beyond its homeostatic role, magnesium exhibits a tight interplay with phosphate metabolism and inflammation. Magnesium and phosphate share intestinal and renal transport pathways, and phosphate can form chelates with Mg^2+^, reducing its bioavailability in conditions of high phosphate load. Conversely, magnesium deficiency enhances phosphate-induced oxidative stress, vascular calcification, and endothelial dysfunction. This reciprocal relationship is particularly relevant in CKD, where both elevated phosphate and low magnesium levels promote oxidative and inflammatory injury. In vitro, phosphate overload promotes intracellular ionic imbalance and mitochondrial dysfunction through Mg-ATP extrusion and ROS overproduction, linking phosphate toxicity to mitochondrial oxidative stress [18]. These mechanisms may underlie the pro-calcifying and pro-senescent effects of phosphate at the cellular level.

Magnesium exerts protective actions against these phosphate-driven mechanisms. Experimental studies have shown that magnesium supplementation mitigates phosphate-induced vascular calcification and inflammation, partly through the inhibition of NF-κB activation and oxidative stress and by suppressing the Wnt/β-catenin pathway involved in osteogenic transdifferentiation of vascular smooth muscle cells [43,44]. Furthermore, magnesium serves as a cofactor for key antioxidant enzymes, including superoxide dismutase and glutathione peroxidase, enhancing cellular resistance to phosphate-induced ROS generation [18].

Clinically, higher serum magnesium concentrations within the normal range are associated with lower cardiovascular mortality and reduced inflammation in CKD patients, independent of albumin or C-reactive protein [45]. Taken together, these findings position magnesium as a counter-regulatory element within the phosphate–inflammation–oxidative stress axis, contributing to vascular protection and improved outcomes in CKD.

Hypomagnesemia (<1.7 mg/dL) affects 10–30% of patients with diabetes and up to 60% of hospitalized individuals, and is associated with increased cardiovascular mortality, arrhythmias, hypertension, and insulin resistance. Deficiency enhances renal potassium and calcium loss, promotes vasoconstriction, and reduces nitric oxide bioavailability, contributing to endothelial dysfunction [42].

##### Implications of Magnesium Abundance in CKD and Phosphocalcic Metabolism

In CKD, magnesium exerts a protective role against vascular calcification and fibrosis. It inhibits phosphate uptake through PiT-1, suppresses Runx2 and BMP-2 expression in VSMCs, and mitigates osteogenic transdifferentiation of VSMCs [36]. Additionally, magnesium modulates PTH secretion by interacting with the CaSR, helping to maintain bone homeostasis. Experimental studies show that magnesium decreases the expression of DKK1 and SFRP1, antagonists of the Wnt/β-catenin pathway, potentially restoring physiological Wnt signaling in endothelium and bone [36,43].

The MagicalBone clinical trial [46] demonstrated in patients with CKD stages 3–4 that supplementation with oral magnesium carbonate (360 mg/day for 15 months) improved arterial stiffness by decreasing pulse wave velocity and without inducing hypermagnesemia or altering bone metabolism. Magnesium-treated patients showed increased urinary magnesium excretion and significant reductions in serum DKK1 and SFRP1 levels, suggesting a beneficial modulation of the Wnt/β-catenin axis and therefore a potential protection against vascular calcification. These effects translated into improved endothelial function and reduced systemic oxidative stress [46].

Other clinical and experimental studies confirm that magnesium supplementation not only prevents but can also reverse vascular calcification and reduce mortality in experimental uremic models [36], alongside improvements in blood pressure and renal function. In humans, low serum magnesium predicts greater arterial stiffness and cardiovascular events, whereas normal magnesium levels are associated with improved metabolic and renal outcomes [46].

##### Therapeutic Implications

Magnesium should be regarded as a pleiotropic agent in CKD with antioxidant, anti-inflammatory, and anti-calcification properties. Regular monitoring of serum magnesium, and when possible, urinary magnesium excretion, provides valuable information on the risk of calcification and CKD progression.

Organic formulations (citrate, lactate, or aspartate) show better bioavailability than inorganic forms (oxide or carbonate) [42]. Altogether, magnesium emerges not only as a marker of mineral balance but also as a potential therapeutic modulator of the phosphate–Wnt–oxidative axis [46]. Further, magnesium is an accessible parameter in clinical practice, without entailing an increase in expenses. Thus, on the basis of the abovementioned findings, magnesium salts might be considered as safe supplements in people with normal renal function and in CKD. Magnesium supplementation, although generally safe, may cause gastrointestinal intolerance and carries a risk of hypermagnesemia in advanced CKD, requiring careful monitoring in some patients.

#### 4.4.2. Zinc

Zinc inhibits the NF-κB pathway through induction of TNFAIP3, thereby reducing vascular osteogenesis and endothelial inflammation. Zinc deficiency, common in CKD and dialysis patients, is associated with increased arterial stiffness and cardiovascular events. Correction of this deficiency restores insulin sensitivity, improves endothelial function, and exerts systemic antioxidant effects [25]. Nevertheless zinc supplementation must be used cautiously, as excessive zinc intake induces intestinal metallothionein expression, which preferentially binds copper, potentially leading to copper deficiency, anemia, and neurotoxicity [47].

#### 4.4.3. Klotho and Fetuin-A

Increasing Klotho and fetuin-A levels might represent an indirect therapeutic goal. Both Klotho and fetuin-A act as endogenous inhibitors of calcification. Klotho antagonizes Wnt/β-catenin signaling and mitigates renal fibrosis [20,35], while fetuin-A forms soluble calcium–phosphate complexes that prevent tissue precipitation and activation of inflammatory pathways [48]. In Klotho-deficient models, normalization of Klotho reduces interstitial fibrosis, inflammation, and vascular stiffness [17,35] (Figure 3).

### 4.5. Antioxidants and Oxidative Stress Modulators

Modulating endogenous antioxidant and metabolic pathways such as AMPK and SIRT1 provides a broader therapeutic framework, particularly because phosphate excess is a major trigger of oxidative stress, mitochondrial dysfunction, and endothelial injury in CKD. High phosphate suppresses eNOS-derived NO, enhances NOX-dependent ROS generation [8], and promotes endothelial senescence [49], creating a metabolic environment in which AMPK and SIRT1 become critically impaired. Activation of AMPK, via agents such as AICAR or metformin, has been shown to restore energy homeostasis and attenuate renal oxidative and inflammatory responses in experimental CKD. Likewise, SIRT1 improves mitochondrial function, downregulates pro-oxidant pathways, and supports endothelial integrity. Collectively, reinforcing AMPK and SIRT1 activity may counteract the vascular toxicity associated with chronic phosphate overload [50].

### 4.6. Emerging Strategies: miRNAs, Extracellular Vesicles, and Wnt/NOX Signaling

MicroRNAs are fine-tuned regulators of gene expression involved in calcification and inflammation. Among them, miR-125b has emerged as a negative regulator of Runx2, and its downregulation is associated with progression of vascular calcification [7]. This finding opens the door to using miRNAs not only as biomarkers but also as potential therapeutic agents.

Moreover, the development of specific NOX inhibitors and modulators of Wnt/β-catenin and TGF-β pathways offers promising opportunities to slow fibrosis and vascular calcification progression in preclinical models [20,51].

Despite the promising mechanistic and early clinical data supporting these interventions, several limitations must be acknowledged. Evidence for melatonin and Tempol remains predominantly preclinical, with no large randomized controlled trials evaluating cardiovascular or renal endpoints in humans. Similarly, while AMPK/SIRT1 activation represents an attractive metabolic target, current pharmacological activators have important restrictions: metformin is limited by the risk of lactic acidosis in advanced CKD, and agents such as AICAR have not been approved for clinical use. Collectively, these considerations highlight that the therapeutic approaches discussed should be viewed as emerging and investigational, and future clinical trials are necessary to define their true efficacy and safety in CKD.

To provide a structured overview of the current therapeutic evidence, Table 1 summarizes representative preclinical and clinical studies evaluating dietary, pharmacological, and metabolic interventions targeting phosphate overload. The table highlights study design, main findings, and the corresponding level of evidence to facilitate comparison across therapeutic strategies.

## 5. Vascular Calcification and Cardiovascular Remodeling

The chronic inflammatory and oxidative milieu induced by phosphate excess constitutes the priming event for vascular calcification (VC), a hallmark complication in CKD and a strong predictor of cardiovascular events and mortality [28,34,57]. Unlike classical atherosclerosis, CKD-related VC predominantly affects the tunica media, driven by the osteogenic transdifferentiation of VSMC under uremic stimuli such as phosphate, calcium, ROS, and inflammatory cytokines [25,26,28].

### 5.1. Cellular and Molecular Mechanisms

As detailed in Section 3 and illustrated in Figure 2, phosphate uptake through the PiT-1 cotransporter activates the osteogenic program in VSMCs [26,58]. High calcium on the other hand, favors apoptosis and the release of membrane-derived vesicles loaded with hydroxyapatite crystals, which act as mineralization nuclei [28,34].

In addition, elevated extracellular calcium alters intracellular homeostasis in VSMC, promoting the formation of matrix vesicles depleted of mineralization inhibitors such as matrix Gla protein and enriched in annexin A6 and phosphatidylserine, which together serve as nucleation complexes for hydroxyapatite deposition and are considered an early pathological trigger of calcification [59].

Recent studies demonstrate that phosphate burden also activates inflammatory cell-death programs in VSMCs. Specifically, high phosphate triggers a NLRP3–caspase-1–gasdermin D–dependent pyroptotic pathway, leading to IL-1β-independent cell lysis and the extracellular release of ASC and caspase-1, which function as pro-mineralizing signals [60]. This response is amplified by potassium efflux through Kir channels, which promotes calcium influx and further NLRP3 activation a mechanism distinct from canonical P2X7 signaling in immune cells [60].

In parallel, calciprotein particles (CPPs) formed by calcium–phosphate precipitation are readily internalized by endothelial cells and VSMCs. Within the endothelium, CPPs trigger oxidative and inflammatory responses and release paracrine mediators that exacerbate VSMC calcification [48,58,61]. Together, these findings identify phosphate and calcium overload as synergistic drivers of vascular mineralization, acting through metabolic dysfunction, inflammatory signaling, and programmed cell-death pathways.

### 5.2. Protective Modulators

Disarrangements of promoters of VC require the downregulation of factors that halt or reduce the development of VC.

Among emerging endogenous inhibitors, the vasoconstriction-inhibiting factor (VIF), a chromogranin A-derived peptide, has been identified as a potent calcimimetic of the CaSR. VIF reduces vascular smooth-muscle calcification by decreasing calcium influx, oxidative stress, and IL-6 secretion, while enhancing the production of carboxylated matrix Gla protein, a key mineralization inhibitor. In experimental CKD models, VIF treatment markedly reduced medial calcification and arterial stiffness, positioning this peptide as a promising therapeutic target for vascular protection [62].

Fetuin-A, a liver-derived glycoprotein that binds to Calcium–Phosphate to form stable CPPs, which prevents the precipitation on different tissues at the time that exerts anti-inflammatory and anti-apoptotic effects in VSMCs [48]. Recent clinical evidence indicates that lower fetuin-A concentrations independently predict new cardiovascular events in patients on hemodialysis, whereas shorter serum T50a marker of calcification propensity better predicts mortality after such events, suggesting complementary but distinct roles in cardiovascular risk modulation [63].

### 5.3. Experimental and Clinical Evidence

In the CRIC study higher serum phosphate independently associated with the presence and severity of coronary calcification, whereas FGF23 showed no consistent association after multivariate adjustment [57]. However, findings across the literature are heterogeneous. Several large population-based and CKD cohorts have identified FGF23, rather than serum phosphate, as the stronger predictor of cardiovascular outcomes, mortality, or LVH. Together, these data suggest that phosphate and FGF23 reflect complementary, rather than competing, dimensions of disordered mineral metabolism and cardiovascular risk [64].

In murine models high-phosphate diets induce medial arterial calcification, systemic inflammation, and malnutrition, findings prevented by phosphate binders such as lanthanum carbonate [16].

### 5.4. Cardiovascular Remodeling

The impact of phosphate overload and FGF23 extends beyond the vasculature. On the one hand, phosphate indirectly promote LVH, through FGF23 which activates FGFRs on cardiomyocytes via the calcineurin–NFAT pathway, independent of Klotho [2]. On the other, recent evidence showed that phosphate itself can directly induce mitochondrial dysfunction and metabolic remodeling in cardiomyocytes, characterized by impaired oxidative phosphorylation, reduced PGC1α expression, and a shift toward glycolytic metabolism, changes that contribute to cardiac hypertrophy and failure in CKD models [65].

In summary, VC in CKD reflects an imbalance between pro-calcific drivers (phosphate, calcium, ROS, TGF-β/Wnt, CPPs) and inhibitors (fetuin-A, magnesium, zinc, Klotho, miRNAs). This process is tightly linked to cardiovascular remodeling particularly, arterial stiffness and LVH, thereby upholding the connection between mineral dysregulation and cardiovascular mortality.

## 6. Phosphate Alterations and Energy Metabolism

Phosphate is a central substrate in cellular bioenergetics, being essential for ATP synthesis, glycolysis, and mitochondrial oxidative phosphorylation. Under physiological conditions, phosphate availability tightly couples energy production with cellular demand through ATP–ADP recycling and mitochondrial phosphate transporters. However, in CKD, sustained phosphate retention disrupts this balance, leading to mitochondrial dysfunction, oxidative stress, and metabolic inflexibility [32].

Experimental studies show that phosphate overload impairs mitochondrial oxidative capacity, increases proton leak, and suppresses the transcriptional coactivator PGC-1α, a master regulator of mitochondrial biogenesis. This process is accompanied by reduced activation of AMPK and SIRT1, two energy-sensing enzymes that normally promote fatty acid oxidation, enhance mitochondrial turnover, and inhibit NF-κB–dependent inflammation [32].

In phosphate-overloaded states, the imbalance between ATP generation and ROS detoxification promotes a “pseudo-hypoxic” phenotype characterized by increased glycolysis and reduced oxidative phosphorylation, similar to metabolic remodeling described in heart failure and diabetic vasculopathy [66]. Similarly, it has been shown that systemic phosphate excess induces malnutrition and energy wasting through chronic inflammation and mitochondrial oxidative stress, linking phosphate toxicity to impaired energy homeostasis at both cellular and organismal levels [6].

Altogether, phosphate retention represents a dual threat to metabolic health. Phosphate not only promotes vascular and skeletal pathology but also drives cellular energy depletion and redox imbalance through AMPK–SIRT1 inactivation. These findings highlight phosphate as a modifiable determinant of metabolic aging and suggest that strategies restoring phosphate balance may help recover mitochondrial efficiency and energy homeostasis in CKD and beyond [13,66].

Disruption of phosphate-dependent energy homeostasis may have direct clinical implications in CKD. Mitochondrial dysfunction and impaired AMPK–SIRT1 signaling contribute not only to reduced oxidative phosphorylation but also to systemic energy deficit, manifesting as muscle wasting, fatigue, and frailty, hallmarks of the MIA syndrome. These findings suggest that phosphate-driven energetic stress extends beyond cellular metabolism to influence whole-body catabolic state and functional decline in CKD patients [13,32].

## 7. Conclusions, Clinical Implications and Future Perspectives

### 7.1. Conclusions

The evidence summarized in this review supports the concept that phosphate is not merely a biochemical marker but an upstream trigger of oxidative stress and inflammatory pathways. In CKD, this leads to on vascular calcification, arterial stiffness, and LVH. These effects are orchestrated through the interaction of the FGF23–Klotho–PTH axis with key signaling cascades such as NOX/ROS, Wnt/β-catenin, and TGF-β, and are further amplified by CPPs and endothelial dysfunction.

In parallel, phosphate excess contributes to MIA syndrome, impacting haematopoiesis, energy metabolism, and overall prognosis. Altogether, these findings underscore that controlling total phosphate burden (rather than serum phosphate alone) and modulating oxidative stress are central strategies to improve cardiovascular and renal outcomes in all subjects, although with greater attention to CKD patients.

### 7.2. Clinical Implications

An effective management approach must be multidimensional, integrating:Dietary restriction focused on inorganic phosphate, guided by the urinary P/UUN ratio; both in subjects with normal renal function and with CKD.Non–calcium-based phosphate binders, as early as possible, to reduce intestinal absorption and lower FGF23.Magnesium and zinc as pleiotropic anti-calcific and anti-inflammatory modulators.Antioxidant and metabolic modulators (AMPK/SIRT1 activators) to restore redox and mitochondrial homeostasis.Emerging therapies including microRNAs, extracellular vesicles, and inhibitors of NOX and Wnt/TGF-β pathways with promising translational potential.

Therapeutic choices should be individualized according to CKD stage, vascular phenotype (medial vs. intimal calcification), metabolic comorbidities, and indicators of phosphate load.

### 7.3. Future Perspectives

Research priorities may include:Randomized clinical trials combining phosphate control (reduced phosphate intake and binders) with magnesium and/or antioxidants, using hard outcomes (cardiovascular/renal mortality) alongside intermediate endpoints (pulse wave velocity, CT calcification scores, ROS biomarkers, and miRNAs).Validation of the urinary P/UUN ratio as a tool for personalized medicine, integrating it into nutritional decision-making algorithms.Development and clinical testing of selective NOX inhibitors and Wnt/TGF-β modulators, defining safety profiles, tissue targets, and therapeutic windows in CKD.Translational mechanistic studies on CPPs and endothelial–VSMC crosstalk, aiming to transform their quantification or neutralization into stratification and therapeutic strategies.Integration of emerging biomarkers (e.g., miR-125b) with vascular imaging and functional metrics, complemented by multi-omic and AI-based approaches to phenotype risk and predict therapeutic response.

In summary, the combination of early phosphate control, redox modulation, and adjunct therapies targeting key signaling axes provides a rational pathway to decouple CKD from its most lethal cardiovascular complications, paving the way toward precision medicine in mineral and bone metabolism.

## Figures and Tables

**Figure 1 ijms-27-00408-f001:**
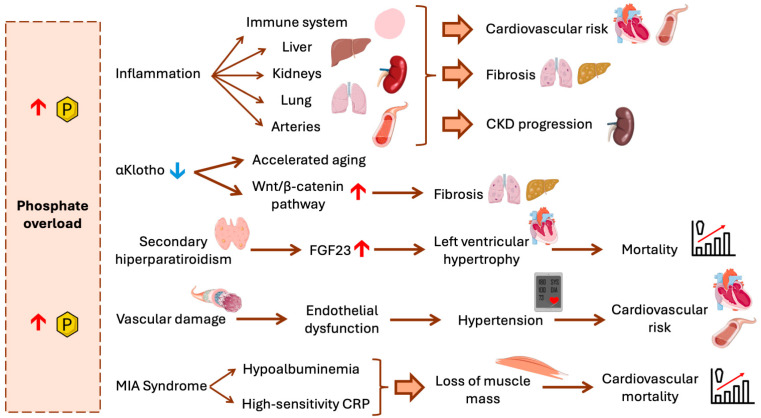
Systemic Pathophysiological Effects of Phosphate Overload and Its Clinical Consequences. Elevated phosphate levels induce damage at multiple physiological levels. They trigger multi-organ inflammation affecting the immune system, liver, kidneys, lungs, and vasculature, all of which contribute to increased cardiovascular risk, soft-tissue fibrosis, and progression of CKD. As CKD advances, α-Klotho levels decline, accelerating aging processes and promoting activation of the Wnt/β-catenin signaling pathway, ultimately leading to organ fibrosis. High phosphate levels also induce secondary hyperparathyroidism, which increases FGF23 concentrations and contributes to left ventricular hypertrophy and higher mortality rates. Additionally, vascular injury caused by phosphate toxicity results in endothelial dysfunction, hypertension, and a further increase in cardiovascular risk. In parallel, the MIA syndrome is characterized by hypoalbuminemia and elevated high-sensitivity C-reactive protein, promoting loss of muscle mass and further increasing cardiovascular mortality. CKD: Chronic kidney disease; MIA: malnutrition–inflammation–atherosclerosis.

**Figure 2 ijms-27-00408-f002:**
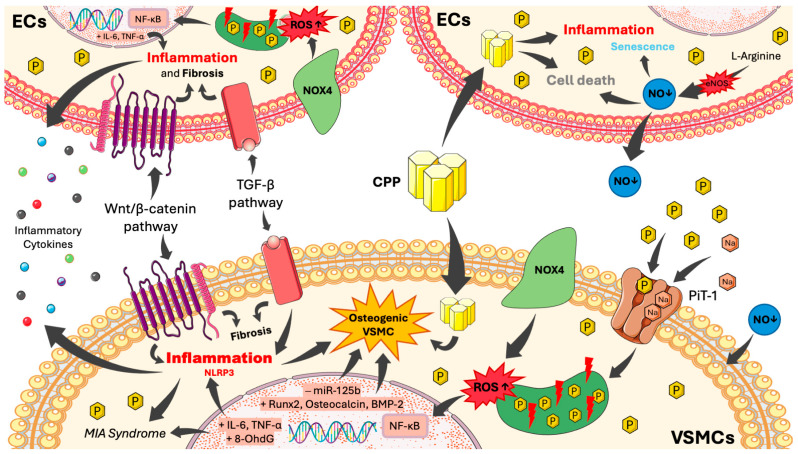
Phosphate-induced cellular damage mechanisms. In VSMCs, extracellular phosphate overload promotes phosphate influx through the PiT-1 cotransporter, triggering mitochondrial dysfunction, NOX4-derived ROS production, and activation of pro-inflammatory (NLRP3 inflammasome) and profibrotic pathways (NF-κB, TGF-β, and Wnt/β-catenin). These events collectively result in cytokine release. The resulting inflammatory mi-lieu, together with increased expression of Runx2, osteocalcin, and BMP-2, and decreased levels of miR-125b, drives the acquisition of an osteogenic phenotype, which is further exacerbated by the formation of CPPs. In ECs, high phosphate concentrations similarly induce oxidative stress, mitochondrial dysfunction, NOX4-derived ROS production, inflammation, and fibrosis through CPPs and the NF-κB, TGF-β, and Wnt/β-catenin pathways. Excess phosphate also induces cell death and senescence through CPP formation and eNOS uncoupling. VSMCs: Vascular Smooth Muscle Cells; PiT-1: Sodium-dependent phosphate cotransporter 1; NOX4: NADPH Oxidase 4; ROS: Reactive Oxygen Species; NLRP3: NOD-, LRR- and pyrin domain-containing protein 3; NF-κB: Nuclear Factor kappa B; TGF-β: Transforming Growth Factor Beta; MIA: Malnutrition–Inflammation–Atherosclerosis; Runx2: Runt-related transcription factor 2; BMP-2: Bone Morphogenetic Protein 2; CPPs: Calciprotein Particles; ECs: Endothelial Cells; eNOS: Endothelial Nitric Oxide Synthase; CKD: Chronic Kidney Disease.

**Table 1 ijms-27-00408-t001:** Summary of preclinical and clinical evidence supporting therapeutic strategies targeting phosphate overload.

Therapeutic Strategy	Representative Studies	Study Design	Main Findings	Level of Evidence
Dietary phosphate restriction/reduction in inorganic additives [10,11]	Pendón-Ruiz de Mier et al., 2021; Novillo et al., 2025	Observational cohorts; Human controlled dietary interventions.	Reduced inorganic phosphate load, decreased P/UUN ratio, attenuated postprandial phosphate and FGF23 rise	Moderate (clinical + mechanistic)
Non–calcium-based phosphate binders (sevelamer, lanthanum) [52,53,54]	Habbous et al., 2017 (NDT meta-analysis, 51 RCTs, 8829 patients); (RCT); Chertow et al., 2002; Block et al., 2005 (RCT);	Systematic review + randomized clinical trials	Sevelamer reduces hypercalcemia (RR 0.27), hospitalizations (RR 0.50), and slows CAC progression (–101 Agatston units). Trend toward lower mortality vs. calcium binders. Lanthanum similar efficacy with fewer hypercalcemic events.	High (multiple RCTs + meta-analysis)
Magnesium supplementation [43,44,45,46,55]	Díaz-Tocados et al., 2017; Rodrigo López-Baltanas et al. 2021; Cayetana Moyano-Peregrin et al., 2025; Pendón-Ruiz de Mier et al., 2024 (MagicalBone Trial).	Observational cohorts + preclinical models + pilot human trial in CKD stages 3–4	Magnesium inhibits phosphate-induced osteogenic signaling, reduces Wnt/β-catenina and Runx2/BMP-2 activation, attenuates vascular calcification in preclinical models, and improves arterial stiffness in early human trials	Moderate to High (robust preclinical + emerging clinical evidence)
Zinc supplementation [25,56]	Voelkl et al., 2019 experimental VSMC, Voelkl et al., 2018 mouse CKD models.	In vitro + in vivo preclinical	Zinc inhibits NF-κB activation, reduces oxidative stress, suppresses osteogenic differentiation, and attenuates vascular calcification in CKD mice.	Moderate (solid preclinical evidence)
Tempol (superoxide dismutase mimetic) [37]	Yamada et al., 2014	Animal CKD/calcification models	Reduced NOX4, 8-OHdG, Runx2 and osteocalcin; attenuated vascular calcification without lowering serum phosphate	Low (preclinical only)
AMPK/SIRT1 activation (AICAR, metformin) [32,50]	Hallows et al., 2010; Dugan et al., 2013;	Preclinical studies + limited clinical observations	Improved mitochondrial biogenesis and energy metabolism; reduced oxidative stress and inflammation; metformin limited in advanced CKD	Low to Moderate (mainly preclinical)

## Data Availability

No new data were created or analyzed in this study. Data sharing is not applicable to this article.

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
