# Peer review of "Phosphate and Inflammation in Health and Kidney Disease"

_ijms, 2025, doi:10.3390/ijms27010408_

Round 1
Reviewer 1 Report
Comments and Suggestions for Authors
This is a valuable and conceptually strong review that addresses a critically important topic in nephrology and cardiovascular medicine. The proposed "phosphatopathy" framework is compelling.
1.The term "Phosphatopathy" is introduced as a key concept but is not formally defined until the end of the introduction. To strengthen the manuscript's thesis, this term should be clearly and concisely defined earlier, ideally in the Abstract or the very beginning of the Introduction.
2.The conceptual framework linking phosphate overload to the MIA syndrome, vascular calcification, and energy disaster is clear. However, a graphical abstract or a central summary figure illustrating this integrative pathophysiology—from phosphate intake to end-organ damage through the various described pathways—would be immensely beneficial.
3.Sections 3 (Inflammation), 4 (MIA Syndrome), and 5 (Vascular Calcification) contain considerable thematic overlap (e.g., NF-κB, ROS, VSMC osteogenesis). This risks redundancy. Using Figure 1 more explicitly in the text to avoid repeating molecular pathways.
4.The claim that “serum phosphate independently associates with coronary calcification” (line 474, citing CRIC study) is presented without acknowledging conflicting literature. Some large cohorts show FGF23—not phosphate—as the stronger predictor of CV outcomes.
5. The review enthusiastically endorses Mg²⁺, Zn²⁺, melatonin, Tempol, and AMPK/SIRT1 activators—but lacks discussion of limitations, side effects, or failed trials.
6. Figure 1 is excellent but dense. Consider splitting into two panels: (A) cellular mechanisms in VSMCs/ECs; (B) systemic consequences (MIA, LVH, etc.).
7. Figure 2 lists many agents (DKK1, SFRP1, AICAR) without explaining their clinical availability. Add footnotes indicating which are experimental vs. clinically used.
Author Response
Responses to Reviewers regarding the manuscript “IJMS-4014730 Phosphate and inflammation in health and kidney disease” R1
Reviewer 1 – Comment 1
“The term "Phosphatopathy" is introduced as a key concept but is not formally defined until the end of the introduction. To strengthen the manuscript's thesis, this term should be clearly and concisely defined earlier, ideally in the Abstract or the very beginning of the Introduction.”
Response
We thank the reviewer for this helpful comment. We agree that the term Phosphatopathy is central to the message of the review and should be defined early for the reader. In response, we have now added a concise definition of Phosphatopathy in the Abstract, so that the concept is clearly introduced from the outset (Abstract, lines 19-22). The more detailed description of Phosphatopathy has been retained in the Introduction, where it appears within the broader pathophysiological framework of phosphate overload and CKD. We believe that this two-step presentation improves the clarity and coherence of the manuscript and addresses the reviewer’s concern.
Reviewer 1 – Comment 2
“The conceptual framework linking phosphate overload to the MIA syndrome, vascular calcification, and energy disaster is clear. However, a graphical abstract or a central summary figure illustrating this integrative pathophysiology—from phosphate intake to end-organ damage through the various described pathways—would be immensely beneficial.”
Responce
We deeply appreciate the comment from the reviewer. In this line, we include a new figure, now referenced as figure 1 to resume the Systemic Pathophysiological Effects of Phosphate Overload and Its Clinical Consequences, including the MIA syndrome, VC and some other of the alterations associated with phosphate overload.
Reviewer 1 – Comment 3:
“Sections 3 (Inflammation), 4 (MIA Syndrome), and 5 (Vascular Calcification) contain considerable thematic overlap (e.g., NF-κB, ROS, VSMC osteogenesis). This risks redundancy. Using Figure 1 more explicitly in the text to avoid repeating molecular pathways.”
Response:
We thank the reviewer for this insightful observation. We agree that Sections 3, 4, and 5 share several upstream molecular pathways, including NOX/ROS signalling, NF-κB activation, PiT-1–mediated phosphate uptake, and early osteogenic transdifferentiation, which created unnecessary repetition in the original version.
To address this, we have revised the text in all three sections to reduce redundancy and to more clearly differentiate their roles:
Section 3 (Inflammation) now provides the core mechanistic framework.
Sections 4 and 5 were edited so that repeated molecular descriptions were removed and replaced by explicit references to Section 3 and to Figure 1 (Now figure 2) as the central summary of the shared pathways.
Transitional phrases were added to clarify how these upstream mechanisms diverge into the MIA phenotype or into vascular calcification.
Several paragraphs were streamlined (e.g., those describing VSMC osteogenic signalling, oxidative stress effects, and cell-death pathways) to improve cohesion and avoid mechanistic duplication.
We believe these changes substantially improve readability, strengthen the logical flow between sections, and enhance the integrative role of Figure 1 as the unifying mechanistic reference.
Reviewer 1, Comment 4:
“The claim that “serum phosphate independently associates with coronary calcification” (line 474, citing CRIC study) is presented without acknowledging conflicting literature. Some large cohorts show FGF23—not phosphate—as the stronger predictor of CV outcomes”.
We thank the reviewer for this important observation. We agree that the literature contains heterogeneous findings regarding the relative predictive value of serum phosphate versus FGF23 for cardiovascular outcomes.
To address this, we have now revised the text to acknowledge that while some studies (including specific analyses from the CRIC cohort) report an independent association between serum phosphate and coronary calcification, other large cohorts identify FGF23 as a stronger predictor of cardiovascular risk (lines 543-547).
We have incorporated additional references and rephrased the statement to provide a balanced interpretation consistent with the broader evidence.
Reviewer 1 – Comment 5:
“The review enthusiastically endorses Mg²⁺, Zn²⁺, melatonin, Tempol, and AMPK/SIRT1 activators—but lacks discussion of limitations, side effects, or failed trials.”
Response:
We thank the reviewer for this valuable comment. We fully agree that the therapeutic section should present a more balanced and critical view of the available evidence. In response, we have incorporated a detailed discussion of the limitations and safety considerations associated with each therapeutic strategy.
Specifically:
Magnesium: We added a paragraph describing the risk of gastrointestinal intolerance and hypermagnesemia in CKD, the need for monitoring (Lines 428-430).
Zinc: We included a detailed explanation of the risk of copper deficiency due to zinc-induced metallothionein up-regulation, as well as the lack of long-term safety data in CKD (Lines 439-442).
Melatonin and Tempol: We clarified that current evidence is predominantly preclinical and that no large randomized controlled trials have demonstrated cardiovascular or renal benefits in humans (Lines 474-481).
AMPK/SIRT1 activators: We noted the clinical limitations of existing agents (e.g., metformin use restricted in advanced CKD because of lactic acidosis risk; AICAR not approved for clinical application (Lines 474-481).
Additionally, we added a concluding paragraph summarizing the overall limitations of these emerging therapies and emphasizing the need for controlled clinical trials to establish efficacy and safety
We believe these revisions provide a more rigorous and balanced assessment, fully addressing the reviewer’s concern.
Reviewer 1 – Comment 6:
- Figure 1 is excellent but dense. Consider splitting into two panels: (A) cellular mechanisms in VSMCs/ECs; (B) systemic consequences (MIA, LVH, etc.).
We appreciate the reviewer’s comment and we apologize about the density of the Figure 1. Therefore, the original Figure 1 has been split into two separate figures. New Figure 1 gather pathologies and systemic consequences of phosphate overload. New Figure 2 focuses on the cellular mechanism in VSMCs and ECs. We also apologize for retaining some terms that we consider highly important at the cellular level.
Reviewer 1 – Comment 7:
- Figure 2 lists many agents (DKK1, SFRP1, AICAR) without explaining their clinical availability. Add footnotes indicating which are experimental vs. clinically used.
We are grateful to the reviewer for the valuable assessment and regret that this information was absent from the original version of the manuscript. Reviewer 2 provided similar constructive feedback, and we are also sorry for the lack of clarity in the previous version. Therefore, we have included a table summarizing the clinical evidence of the agents and therapeutic approaches, indicating whether they correspond to preclinical or clinical studies, along with other significant information.
Reviewer 2 Report
Comments and Suggestions for Authors
This review provides biochemical, molecular, and clinical insights into phosphate metabolism and its role in oxidative stress, inflammation, and vascular injury, particularly in kidney disease. The manuscript is comprehensive and timely. Below are several comments for improvement.
Major Comments
- The review primarily focuses on phosphate overload–related complications. However, in the general population, several studies have reported a U-shaped association between serum phosphate levels and mortality, indicating that both excessively high and low phosphate concentrations may be detrimental to health. Increased renal phosphate loss—whether mediated by FGF23 or due to genetic, neoplastic, or acquired causes—can result in hypophosphatemia. Furthermore, various renal tubular disorders can contribute not only to hypophosphatemia but also to kidney dysfunction. Thus, hypophosphatemia should also be discussed as an important aspect of phosphate-related pathophysiology in kidney disease.
- It is unclear whether the term “phosphatopathy” is widely accepted among researchers and clinicians. If this term is newly proposed or not yet standardized, the authors should provide a clear definition, rationale, and evidence-based discussion to support its use.
- Figures 1 and 2 are excellent and effectively summarize the mechanisms of phosphate-induced vascular and renal injury. However, considering that the review’s title includes the term “health”, it would be appropriate to discuss or at least mention that phosphate may also trigger inflammatory processes in other organs—such as the liver, lung, and immune system—beyond cardiovascular and kidney tissues.
- Although the review highlights potential therapeutic strategies—including dietary phosphate restriction, magnesium and zinc supplementation, and SIRT1/AMPK activation—it does not sufficiently discuss the clinical evidence hierarchy supporting these interventions. To enhance clarity and translational value, it would be beneficial to include a table summarizing key preclinical and clinical studies on these therapeutic approaches, outlining their study design, main findings, and level of evidence.
Minor Comments
- The title “Phosphate and inflammation in health and renal disease” is appropriate, but the term “kidney” is now more commonly used than “renal” in international publications and may improve accessibility.
- The first paragraph of Section 3 (“Role of phosphate excess on promotion of systemic inflammation”), which begins with “Phosphate is an essential element for cellular homeostasis…”, is highly similar to the opening paragraph of the Introduction (“Phosphate is an essential mineral for both cellular and systemic homeostasis…”). Consider merging or rephrasing to reduce redundancy.
- In the References section, citations 5 and 36 appear to reference the same study. Please verify and correct any duplication.
Author Response
Reviewer 2 – Comment 1:
“The review primarily focuses on phosphate overload–related complications… however, hypophosphatemia should also be discussed as an important aspect of phosphate-related pathophysiology.”
Response:
We thank the reviewer for this comment. We agree that disturbances in phosphate homeostasis are bidirectional, and that hypophosphatemia, particularly when driven by renal phosphate wasting, also carries significant clinical implications. In response, we have added a new paragraph at the end of the Introduction acknowledging: the U-shaped relationship between serum phosphate levels and mortality, the main mechanisms leading to renal phosphate wasting, and the systemic consequences of hypophosphatemia.
At the same time, we clarify that the primary aim of this review is to examine the mechanisms and clinical consequences of phosphate overload, which is why hypophosphatemia is not developed in dedicated subsections. We believe that the added paragraph provides the necessary conceptual balance while maintaining the central focus of the manuscript.
Reviewer 2 – Comment 2:
“It is unclear whether the term ‘phosphatopathy’ is widely accepted among researchers and clinicians. If this term is newly proposed or not yet standardized, the authors should provide a clear definition, rationale, and evidence-based discussion to support its use.”
Response:
We thank the reviewer for this thoughtful observation. We agree that phosphatopathy is not yet a standardized clinical term but rather a conceptual framework used to describe the convergent systemic effects of chronic phosphate overload. In response, we have clarified this explicitly in the manuscript.
We introduced a concise and clear definition of phosphatopathy in the Abstract, so that readers understand the concept from the outset (Lines 99-102)
In the Introduction, after presenting the full definition, we added a sentence clarifying that phosphatopathy is not an established diagnostic category, but a descriptive construct intended to integrate the oxidative, inflammatory, vascular, metabolic, and cellular pathways triggered by chronic phosphate excess.
Reviewer2 – Comment:3
“Figures 1 and 2 are excellent… however, considering that the review’s title includes the term ‘health’, it would be appropriate to discuss phosphate-related inflammation in other organs such as liver, lung, and the immune system.”
Response:
We thank the reviewer for this thoughtful suggestion. We agree that phosphate-induced inflammation is not restricted to the cardiovascular and renal systems. However, the primary aim of our review is to explore in depth the mechanisms by which phosphate overload contributes to CKD progression, vascular calcification, and systemic inflammation within the MIA framework. For this reason, a comprehensive analysis of phosphate effects in other organs falls outside the intended scope of the manuscript.
To acknowledge the reviewer’s point while maintaining the thematic focus of the review, we have added a brief statement clarifying that phosphate can also activate inflammatory pathways in other tissues, including liver, lung, and immune cells, even though these mechanisms are not addressed in detail here. This clarification has been incorporated at the beginning of Section 3, where the scope of the inflammatory discussion is defined. Additionally, to better emphasize the impact of phosphate overload on additional organs, we have created a new figure (now Figure 1) that outlines the different systemic consequences of elevated phosphate levels.
Reviewer 2 – Comment 4:
“Although the review highlights potential therapeutic strategies—including dietary phosphate restriction, magnesium and zinc supplementation, and SIRT1/AMPK activation—it does not sufficiently discuss the clinical evidence hierarchy supporting these interventions. To enhance clarity and translational value, it would be beneficial to include a table summarizing key preclinical and clinical studies on these therapeutic approaches, outlining their study design, main findings, and level of evidence.”
Response:
We thank the reviewer for this excellent suggestion. We agree that synthesizing the therapeutic evidence in a structured format improves the clarity and translational relevance of the manuscript. In response, we have added a new table (Table 1) summarizing representative preclinical and clinical studies investigating dietary phosphate restriction, non–calcium-based phosphate binders, magnesium and zinc supplementation, Tempol, and AMPK/SIRT1 activation.
The table includes: the study design (preclinical vs. clinical), the main mechanistic or clinical outcomes, and an assessment of the level of evidence for each therapeutic strategy.
Minor Comments
We thank the reviewer for these helpful minor comments. All suggested revisions have now been implemented. Specifically:
- The term “kidney” has replaced “renal” in the title for improved accessibility.
- The redundant opening paragraph in Section 3 has been rephrased to avoid overlap with the Introduction.
- The duplicate references (citations 5 and 36) have been reviewed and corrected.
Round 2
Reviewer 2 Report
Comments and Suggestions for Authors
I have no further comments on the manuscript.